# Identification of Serum Interleukin-22 as Novel Biomarker in Pulmonary Hypertension: A Translational Study

**DOI:** 10.3390/ijms25073985

**Published:** 2024-04-03

**Authors:** Friederike Klein, Sandesh Dinesh, Desiree Fiedler, Katja Grün, Andrea Schrepper, Jürgen Bogoviku, Laura Bäz, Alexander Pfeil, Daniel Kretzschmar, P. Christian Schulze, Sven Möbius-Winkler, Marcus Franz

**Affiliations:** 1Department of Internal Medicine I, University Hospital Jena, Am Klinikum 1, 07747 Jena, Germany; friederike.klein@med.uni-jena.de (F.K.);; 2Department of Cardiothoracic Surgery, University Hospital Jena, Am Klinikum 1, 07747 Jena, Germany; 3Department of Internal Medicine III, University Hospital Jena, Am Klinikum 1, 07747 Jena, Germany; 4Herz-und Gefäßmedizin Goslar (HUGG), Goslar, Fleischscharren 4, 38640 Goslar, Germany; 5Department of Cardiology, Angiology and Intensive Care Medicine, Cardiovascular Center Rotenburg Klinikum Hersfeld-Rotenburg, Heinz-Meise-Straße 100, 36199 Rotenburg an der Fulda, Germany

**Keywords:** IL-22, pulmonary hypertension, biomarker

## Abstract

Growing evidence suggests the crucial involvement of inflammation in the pathogenesis of pulmonary hypertension (PH). The current study analyzed the expression of interleukin (IL)-17a and IL-22 as potential biomarkers for PH in a preclinical rat model of PH as well as the serum levels in a PH patient collective. PH was induced by monocrotalin (60 mg/kg body weight s.c.) in 10 Sprague Dawley rats (PH) and compared to 6 sham-treated controls (CON) as well as 10 monocrotalin-induced, macitentan-treated rats (PH_MAC). Lung and cardiac tissues were subjected to histological and immunohistochemical analysis for the ILs, and their serum levels were quantified using ELISA. Serum IL levels were also measured in a PH patient cohort. IL-22 expression was significantly increased in the lungs of the PH and PH_MAC groups (*p* = 0.002), whereas increased IL17a expression was demonstrated only in the lungs and RV of the PH (*p* < 0.05) but not the PH_MAC group (*p* = n.s.). The PH group showed elevated serum concentrations for IL-22 (*p* = 0.04) and IL-17a (*p* = 0.008). Compared to the PH group, the PH_MAC group demonstrated a decrease in IL-22 (*p* = 0.021) but not IL17a (*p* = n.s.). In the PH patient collective (*n* = 92), increased serum levels of IL-22 but not IL-17a could be shown (*p* < 0.0001). This elevation remained significant across the different etiological groups (*p* < 0.05). Correlation analysis revealed multiple significant relations between IL-22 and various clinical, laboratory, functional and hemodynamic parameters. IL-22 could serve as a promising inflammatory biomarker of PH with potential value for initial diagnosis, functional classification or even prognosis estimation. Its validation in larger patients’ cohorts regarding outcome and survival data, as well as the probability of promising therapeutic target structures, remains the object of further studies.

## 1. Introduction

Pulmonary hypertension (PH) is a clinically and etiologically heterogeneous disease and is defined hemodynamically by a mean pulmonary artery pressure (mPAP) of >20 mmHg at rest [1]. The main cause of its poor prognosis is the development of a right ventricular dysfunction due to pressure overload and consecutive right heart failure [2]. Based upon the etiology, PH can be differentiated into five clinical groups. Group 1, also referred to as pulmonary arterial hypertension (PAH), contains all hereditary forms and the idiopathic pulmonary arterial hypertension (PAH), and groups 2 and 3 include PH associated with left heart disease (2) and PH associated with chronic lung disease and/or hypoxia (3), respectively. PH associated with pulmonary artery obstruction is classified as group 4 and PH with unclear and/or multifactorial mechanisms is summarized in group 5 [1,3].

Besides vasoconstriction and thrombosis, there is growing evidence suggesting an involvement of inflammatory processes in the pathogenesis of PH [4]. Dendritic cells, macrophages, T-cells, B-cells and mast cells, as the cellular drivers of inflammation, can be found in the perivascular connective tissue in patients with PH [5,6,7]. Inflammatory mediators such as interleukin (IL)-6, IL-1b, several Toll-like receptors, a variety of chemokines and TNF alpha were shown to be involved in the inflammatory response in patients with PH and right heart failure [8,9]. In rat-models as well as in humans, a significant correlation between echocardiographic, hemodynamic and clinical parameters and an increase in circulating and stationary immune cells in the lung and right ventricular tissue were shown recently [10,11].

By real-time RT-PCR gene expression analysis, our research group was able to demonstrate a PH-associated relevant regulation of 17 genes in the lung and 20 genes in the RV in a rat model of experimentally induced PH [10]. Here, amongst others, IL-17a and IL-22 were the most relevantly upregulated genes in the lung tissue.

IL-17a is a pro-inflammatory cytokine that is secreted by T-helper 17 cells and binds to the IL-17 receptor, which is expressed mainly on non-hematopoietic cells such as fibroblasts. By acting in two pathways, either by de novo gene transcription or by stabilizing target mRNA transcripts, IL-17a upregulates expression of inflammatory genes [12]. Unchecked expression of IL-17a leads to autoimmunity and endothelial vascular activation, as seen in various diseases like psoriasis or rheumatoid arthritis [13].

IL-22 is secreted primarily by T-cell subsets like T-helper 17 cells, natural killer cells and innate lymphoid cells [14]. Its effect is mediated on non-immune cells at barrier surfaces like skin, lung tissue or gut mucosa [15] through a heterodimeric receptor consisting of IL-22R and IL-10Rβ [16]. It appears to have a dual role with pro- as well as anti-inflammatory properties. This dual function of IL-22 is, amongst others, regulated by the presence or absence of IL-17a in the tissue. In the presence of IL-17a, IL-22 acts as a promoter of inflammation, and in its absence, IL-22 contributes to the regulation of cell proliferation for repair following injury [17]. In acute inflammations, IL-22 acts by propagating cell proliferation, inhibiting apoptosis and also increasing mucin production. However, these factors lead also to its detrimental effects in chronic inflammation, resulting in hyperplasia [18].

Up to now, specific biomarkers reflecting pulmonary vascular or even the prognostically relevant right ventricular remodeling do not exist but are absolutely needed to increase the quality of diagnostic tests and pave the way for novel therapeutic approaches.

The aim of the present study was, one the one hand, to evaluate the occurrence of the above-mentioned promising interleukins in lung and right ventricular tissue as well as serum in a representative animal model and to correlate the results with clinical parameters such as mPAP and certain echocardiographic right heart dysfunction markers. On the other hand, we evaluated these interleukins in the serum of patients with known PH to determine their diagnostic potential in daily clinical practice in a real-world setting.

## 2. Results

### 2.1. Right Ventricular Function and Right Ventricular Pressure in the Rat Model

To verify the success of the experimental setting, right heart catheterization via the right internal jugular vein to measure the systolic right ventricular pressure was performed.

The mean systolic right ventricular pressure (RVPsys) was significantly increased in the PH group compared to the controls (87.65 ± 18.14 vs. 37.87 ± 15.38, *p* < 0.01). However, under therapy with MAC, there was a significant decrease in the mean right ventricular pressure compared to the untreated PH group (45.62 ± 14.16 vs. 87.65 ± 18.14, *p* < 0.01, Figure 1).

Echocardiographic assessment of the right ventricle showed a significantly reduced TAPSE in the PH in comparison to the CON group (1.71 ± 0.33 vs. 2.38 ± 0.32, *p* = 0.021), with significant improvement under MAC treatment in the PH_MAC group compared to the PH group (1.71 ± 0.33 vs. 2.08 ± 0.30, *p* = 0.032). Additionally, we measured the right atrial (RA) area, which was significantly higher in the PH and PH_MAC groups (45.1 ± 9.6 and 42.8 ± 11.7 mm^2^) than in the controls (28.9 ± 5.1 mm^2^, *p* = 0.015 and 0.018), but PH and PH_MAC did not differ significantly. The basal right ventricular diameter measured 5.67 ± 0.99 mm in the PH group, which was significantly higher than in the controls (3.6 ± 0.39 mm, *p* = 0.003). With 5.11 ± 0.78 mm in the PH-MAC group, basal right ventricular diameter did not differ significantly compared to the PH group (Figure 1).

### 2.2. PH Associated Lung and Right Ventricular Tissue Damage

The lung tissue damage caused by PH was evaluated using the scoring systems introduced previously (Franz 2016). The histological sum-score in both the PH group and the PH_MAC group was significantly higher than in the CON group (PH 8.33 ± 1.50 vs. PH_MAC 5.60 ± 1.43 vs. CON 1.56 ± 1.11, respectively, all *p* = 0.003). However, the histological sum-score in the PH_MAC group was significantly lower than the PH group without treatment (*p* = 0.003, Figure 2).

The severity of right heart tissue damage was assessed by calculation of the above-mentioned semi-quantitative scoring system (Figure 2). The histological changes in the CON group were low, as expected (0.083 ± 0.10). The tissue injury in both the PH and PH_MAC (1.04 ± 0.64 vs. 0.78 ± 0.39, respectively) groups was significantly higher than in the CON group (both *p* = 0.006). Under therapy with MAC, the tissue injury in the PH_MAC group was lower than that of the untreated PH group, without, however, reaching statistical significance.

### 2.3. Immunofluorescence Based Detection of Interleukins 17a and 22 in Lung and Right Ventricular Tissue

#### 2.3.1. Interleukin 17a

No IL-17a expression could be detected in the lung tissue of the controls, but clear tissue deposition was visualized in both the PH and PH_MAC groups, with a significantly higher fluorescence signal seen in the untreated PH group compared to the MAC_PH group (1.29 ± 0.91 and 0.31 ± 0.37, respectively, *p* < 0.05, Figure 3).

In the right ventricular tissue, the controls showed no deposition of IL-17a. In contrast, there was mild expression in the PH group (0.60 ± 0.55) but not in the PH_MAC group, Figure 3).

#### 2.3.2. Interleukin 22

In the control group, no expression of IL-22 could be visualized, whereas in both the PH and the PH_MAC group, a clear tissue deposition of IL-22 could be demonstrated. Using the aforementioned semi-quantitative scoring system, both the PH and PH_MAC groups (1.40 ± 0.55 and 1.10 ± 0.70, respectively) showed significantly higher deposition of IL-22 in comparison to the CON group, in which no deposition could be visualized. No significant difference in the IL-22 deposition between the PH and PH_MAC groups could be detected (Figure 3). In RV, an IL-22 expression could not be detected in either of the experimental groups Fluorescence due to deposition of IL-22 in RV could not be detected in either of the experimental groups.

### 2.4. Serum Concentrations of Interleukins 17a and 22 in the Rat Model Using ELISA

IL-22 could be detected in the serum of all three experimental groups. In comparison to the controls, the concentration in the PH and PH_MAC groups showed a tendency but did not differ significantly (16.64 ± 6.63 pg/mL vs. 32.71 ± 6.81 pg/mL, respectively 21.05 ± 8.11 pg/mL, Figure 4).

The concentrations of IL-17a in both the PH group and the PH_MAC group (32.32 ± 6.23 pg/mL vs. 30.89 and 11.24 pg/mL, respectively) were significantly higher than in the CON group (13.01 ± 2.23 pg/mL, p_CON-PH_ = 0.024 and p_CON-PH_MAC_ = 0.036). However, the serum concentration in the PH_MAC group under treatment remained comparable to the PH group, without a statistically significant difference (Figure 5).

### 2.5. Correlations of Tissue and Circulating Interleukins, Laboratory Values and Clinical Data in the Rat Model

Correlation analyses between hemodynamic as well as echocardiographic parameters and the amount of circulating interleukins revealed multiple significant results (Table 1). Of particular interest was the correlation of the serum levels of IL-17a and -22 to various hemodynamic and functional parameters, as this could form a basis for usage as biomarkers in the future.

Amongst many other relevant results, the serum levels of IL-22 correlated positively with the histological sum-score of the lung (r = 0.73, *p* = 0.001) and negatively with TAPSE (r = 0.73, *p* = 0.001, Figure 6). Meanwhile, the serum levels of IL-17a correlated significantly with the histological sum-score of RV (r = 0.71, *p* = 0.001) and significantly with RA area in echocardiography (r = 0.59, *p* = 0.01, Figure 6).

### 2.6. Human Subjects Clinical Data

We enrolled 92 patients suffering from PH, diagnosed according to the current ESC guidelines by right heart catheterization with the verification of a mean pulmonary arterial pressure of ≥20 mmHg, irrespective of its etiology. All of them were admitted to our outpatient department of the Clinic for Internal Medicine I, Jena University Hospital, Germany. PH subgroups were specified according to current guidelines (Humbert 2022). A total of 30 control patients without relevant coronary artery or valve disease and normal ejection fraction, as well as no signs of PH, were recruited in our outpatient department. All patients also had no malignant disease, active infection or autoimmune disease.

Baseline characteristics of both groups are summarized in Table 2. Echocardiographic measurements of both collectives are summarized in Table 3.

### 2.7. Serum Concentrations of Interleukins 17a and 22 in Humans Using ELISA and Correlations of Circulating Interleukins, Laboratory Values and Clinical Data

For IL-17a, in over 90% of patients, serum concentration was below the kit-specific analytical detection limit in all groups. Therefore, no further analysis could be performed for IL-17a.

For IL-22, patients showed a median serum IL-22 was 14.0 ± 41.1 pg/mL, which was significantly higher compared to the control group, which presented a median serum IL-22 of 3.0 ± 5.9 pg/mL (*p* < 0.001, Figure 7. Interestingly, the increase in IL-22 could be detected in all PH groups, irrespective of the different etiologies (Figure 7), with no significant differences between groups.

ROCanalysis for the discrimination between PH patients and controls revealed an area under the curve (AUC) of 0.848 (Figure 8).

By logistic regression analysis (backward elimination WALD) including a variety of potentially relevant patients’ characteristics (n = 12), IL-22 could be identified to be the only biomarker independently predicting the presence of PH (OR: 1.649; 95% CI: 1.069–2.543; *p* = 0.024).

Between serum IL-22 levels and clinical parameters, a number of significant correlations could be found, such as BNP (r = 0.367, *p* < 0.001), sPAP (r = 0.248, *p* = 0.017), invasive mean PAP (r = 0.420, *p* = 0.006), basal right ventricular diameter (r = 0.393, *p* = 0.004) and right atrial area (r = 0.412, *p* = 0.001) as well as pulmonary vascular resistance (r = 0.368, *p* = 0.017). Inverse correlations could be found for serum IL-22 levels and 6 min walking test (r = −0.269, *p* = 0.047) and TAPSE (r = −0.459, *p* < 0.001, Appendix A).

IL-22 levels were also analyzed regarding the specific functional class groups I-IV, which are an equivalent for the NYHA classification in patients with PH. IL-22 levels did not only differ between controls and patients with PH (*p* < 0.001) but also among the different functional class groups (*p* = 0.02). Patients with functional class IV had the highest IL-22 serum level, with a mean value of 43.32 ± 59.75 pg/mL, whereas patients with functional class I showed a mean IL-22 value of 13.0 ± 6.99 pg/mL (Appendix A).

## 3. Methods

### 3.1. Animal Model of PH

For this study, a rat model of monocrotaline (MCT)-induced PH was used [19]. Therefore, 10- to 12-week-old male Sprague Dawley-Rats, weighing around 300 g each, were obtained from Charles River Laboratories (Sulzfeld, Germany). The rats were allowed a phase of acclimatization for at least 7 days and all experiments were carried out under controlled day–night rhythms and constant temperatures. The rats had continuous access to food and water during the entire study. They were cared for and monitored by experienced staff.

The animal experiment was approved by the State Office for Consumer Protection (TLV Bad Langensalza, Germany, local registration number: 02-004/14). All experiments were carried out in accordance with the National Institute of Health Guidelines for the Care and Use of Laboratory Animals (8th edition) and the European Community Council Directive for the Care and Use of Laboratory Animals of 22 September 2010 (2010/63/EU).

In this experiment, 26 rats were used: 10 rats with MCT-induced PH (PH) and 10 rats with MCT-induced PH treated with the dual endothelin receptor antagonist Macitentan (PH_MAC), as well as 6 healthy (sham-treated) controls (CON). To induce PH, rats received a single dose of MCT (60 mg/kg body weight, 300 µL, Carl Roth, Karlsruhe, Germany) injected subcutaneously. The treatment group (PH_MAC) additionally received a MAC (Macitentan) dose of 15 mg/kg body weight once daily per os from day 14 until day 28.

Sham-treated control rats received an injection of 300 mL NaCl, which was also the carrier substance for the MCT. In order to avoid opportunistic infections and resulting inflammatory changes, all animals received a prophylactic antibiotic treatment with 2.5% enrofloxacin (WDT, Garbsen, Germany) from day 1 to day 14 added to the drinking water. The animals were weighed twice per week and their overall health was assessed using a clinical severity score (CSS) which accounts for spontaneous activity, body posture and reaction to external stimuli [20].

#### 3.1.1. Echocardiographic and Hemodynamic Assessment

Echocardiographic assessment was performed on day 27 after MCT injection in anesthesia with isoflurane (Abbvie Germany GmbH & Co. KG, Wiesbaden, Germany) for a duration of less than 10 min using the Vevo 770 Rodent-Ultrasound-System (Visual Sonic, Canada, 17 MHz probe RMV176) while constantly monitoring body temperature and respiratory rate.

A variety of surrogate markers of right ventricular size, form and function, e.g., right ventricular diameters, right atrial area (RA area) or the tricuspid annular plane systolic excursion (TAPSE), were measured.

On day 28 after the MCT injection, right heart catheterization was carried out via the right internal jugular vein (1.4F micro conductance pressure-volume catheter, model SPR-839; Millar Instruments Inc; PowerLab system, AD Instruments Ltd., Oxford, UK) to measure the systolic right ventricular or pulmonary artery pressure, respectively. All rats were therefore anesthetized with a single dose of 100 mg/kg body weight ketamine (Zoetis Germany GmbH, Berlin, Germany) and 10 mg/kg body weight of 2% Xylazin (Serumwerke Bernburg, Bernburg, Germany).

After completion of the right heart catheterization, the animals were euthanized in deep anesthesia and analgesia. Cardiac blood and the organs were collected after thoracotomy for further evaluations.

#### 3.1.2. Histological Assessment of PH-Induced Lung and Right Ventricular Tissue Damage

Lung and right ventricular tissue were formalin fixed, paraffin embedded and H&E stained. To evaluate histological alterations in right ventricular tissue, sirius red-stained tissue sections were also used.

The histopathological assessment of lung and right ventricular tissue damage was performed using a scoring system previously described by our research group, which entails all important parameters of pulmonary vascular and lung parenchymal damage observable in the MCT rat model [21]. The scoring system for lung tissue includes 5 histological phenomena commonly occurring in PH: percentage of atelectasis area (not detectable = 0 points, <30% of tissue area = 1 point and ≥30% of tissue area = 2 points), percentage of emphysema area (not detectable = 0 points, <30% of tissue area = 1 point and ≥30% of tissue area = 2 points), degree of media hypertrophy of peribronchial arteries (not detectable = 0 points, weak = 1 point, moderate = 2 points and severe = 3 points), presence of perivascular cellular edema of peribronchial arteries (absent = 0 and present = 2 points) and degree of media hypertrophy of small arteries not directly related to bronchi (not detectable = 0 points, weak = 1 point, moderate = 2 points and severe = 3 points). Right ventricular cardiac tissue was evaluated by assessing interstitial cellularity (in particular inflammatory cell infiltration) as well as the extent of interstitial fibrosis. The extent of each parameter was rated semi-quantitatively as follows: 0 = not detectable (absent), 1 = mildly detectable, 2 = moderately detectable and 3 = strongly detectable. In total, score values per tissue sample could reach from 0 to 12 points for lung tissue damage or 0 to 6 points for myocardial (right ventricular) tissue damage, respectively (a higher score value reflects a more severe tissue damage).

The analyses were performed independently and blinded by two experienced scientists.

#### 3.1.3. Immunofluorescence Based Detection of Interleukins in Lung and Right Ventricular Tissue

For immunofluorescence labelling, 4 µm-thick tissue sections were washed for 20 s in ice-cold methanol and were then fixed for 9 min in ice-cold acetone. For the detection of the antigen of interest, 100 μL of primary antibodies was applied to each slide and incubated overnight. As primary antibodies, the Rabbit Anti-Rat IL-22 (bs-2623R, Bios Antibodies, Woburn, MA, USA) and the Mouse Anti-rat IL-17a (sc-374218, Santa Cruz Biotechnology Inc., Dallas, TX, USA) were used. Subsequently, the slides were washed thrice in TBS-tween solution and were then incubated with the secondary antibodies, Cy3-conjugated AffiniPure goat anti-rabbit and Cy3-conjugated AffiniPure donkey anti-Mouse (both Jackson Immunoresearch Laboratories, Inc., West Grove, PA, USA), for 45 min at room temperature. Finally, the slides were washed twice with TBS-tween solution and once with distilled water and were then covered with Vectashield medium containing DAPI.

Semi-quantitative analysis of the protein expression of the fluorescently labeled interleukins of interest was carried out with the Axioplan 2 Imaging fluorescence microscope, using a semi-quantitative scoring system ranging from 0 (nearly no expression) to 3 (high level of expression).

#### 3.1.4. Quantification of Circulating Interleukins by ELISA

The quantitative analysis of IL-22 and IL-17a in rat serum of the three experimental groups was carried out by enzyme linked-immunosorbent assay (ELISA) technique, using the LSBio IL-22 ELISA Kit (LifeSpan BioSciences, Inc., Seattle, WA, USA) and the Abcam Rat IL-17a ELISA Kit (Abcam PLC, Cambridge, UK) according to the instructions of the manufacturer.

### 3.2. Human Subjects

For biomarker analysis in PH patients, 92 patients suffering from PH of different clinical groups were enrolled in the study. These patients were admitted to the outpatient department of the Clinic for Internal Medicine I, Jena University Hospital, Germany, and agreed to participate in our local clinical registry ‘Pulmonary Hypertension’. All patients and control subjects gave written informed consent for participation before inclusion. The local ethics committee of the Medical faculty of the Friedrich Schiller University Jena has approved the study (registration number 4732-03/16), which was conducted in adherence to the Declaration of Helsinki and good clinical practice (GCP) guidelines. PH was diagnosed according to the current ESC guidelines by right heart catheterization with the verification of a mean pulmonary arterial pressure of >20 mmHg. PH subgroups were specified according to the current guidelines [1]. Exclusion criteria were a malignant disease or an active infection. A total of 30 control patients showing increased cardiovascular risk but no evidence of pulmonary hypertension, ischemic or structural heart disease were also recruited in our outpatient department. If necessary, a relevant coronary artery disease was excluded by coronary angiography. They also had no malignant disease, active infection or autoimmune disease.

All patients received a thorough clinical examination and anamnesis, as well as a 6 min walking test and a transthoracic echocardiography.

In the frame of clinically indicated laboratory diagnostics, blood samples were withdrawn according to local standard operating procedures. Collection tubes were centrifuged within 20 min and serum was transferred into special low binding tubes (Protein LoBind, Eppendorf AG, Hamburg, Germany), which were stored at −80 °C until further analysis.

#### Assessment of Circulating Interleukins by ELISA in Human Subjects

The quantitative analysis of IL-22 and IL-17a in serum of the two patient groups was carried out by ELISA, using the sandwich technique with commercially available kits Quantikine ELISA Human IL-17 and Quantikine ELISA Human IL-22 (R&D Systems, Minneapolis, MN, USA) according to the instructions of the manufacturer.

### 3.3. Statistical Analysis

Data are expressed as mean and standard deviation and were calculated using Microsoft Excel. The software IBM SPSS statistics, version 27.0 (IBM Inc., Armonk, NY, USA) was used for the statistical analysis. As the number of experimental animals is limited, an assumption of normal distribution of the data is not possible. Hence, non-parametric tests in the statistical analysis were used. The Mann–Whitney U Test was used to test for significance of the analyzed parameters between the different experimental groups. Bi-variate correlations between non-parametric variables were tested using Spearman’s correlation coefficient. The significance level was set at *p* ≤ 0.05. When comparing multiple groups, *p*-values were corrected using the Bonferroni–Holm method. For the evaluation of parameters showing significant differences between PH patients and controls with respect to their predictive value for the presence of PH, we performed multivariate regression analysis by using a binary logistic model (backward elimination method: Wald). Therefore, the whole study cohort was applied and, due to the limited number of patients, we did not split them into a discovery and validation cohort. As covariates (*n* = 12), age, systolic blood pressure, diastolic blood pressure, heart rate, functional class, brain natriuretic peptide, C reactive protein, creatinine, LDL cholesterol, hemoglobin, platelet count and IL-22 were included in the model. Moreover, for IL-22, receiver operating characteristic (ROC) analysis was carried out to distinguish between PH patients and controls.

## 4. Discussion

In clinical practice, PH is often diagnosed much too late, which is due to the non-specific clinical presentation. Thus, patients frequently already show an advanced stage of the disease at initial diagnosis [22]. Earlier diagnosis and following treatment initiation are crucial regarding prevention of disease progression and the long-term outcome of PH patients. Laboratory biomarkers could help in this process by providing relatively easy means to distinguish possible etiologies for patients’ symptoms, leading to timely diagnosis of PH.

In this translational study, we were able to demonstrate increased expression of IL-17a in lung and right ventricular tissue in a preclinical rat model of PH, which was attenuated by treatment with MAC. For IL-22, tissue expression was increased in the lung but not the RV of PH rats compared to controls. There was no significant attenuation in case of MAC treatment. Serum IL-17a and IL-22 levels were elevated in untreated PH rats compared to controls. In contrast to serum IL-22, which was decreased in case of MAC treatment, IL-17a in serum did not show a treatment response.

The increase in serum IL-22 could also be shown in the collective of PH patients, regardless of its etiology, compared to a healthy control group. Moreover, IL-22 serum levels did not differ between the PH groups. However, IL-17a could not be detected in the human serum.

The increased IL-17a expression in lung and RV tissue shown in this study is consistent with the results of previous work of our study group, in which IL-17a was amongst the most upregulated genes in lung tissue in a rat model of experimentally induced PH [10]. IL-17a is secreted by T-helper 17 cells and binds to the IL-17 receptor that is mainly expressed by non-hemopoietic cells such as fibroblasts [12,23]. As a pro-inflammatory cytokine, it upregulates the expression of inflammatory genes and can therefore lead to autoimmunity and endothelial vascular activation, as was shown for various diseases, e.g., psoriasis, rheumatoid arthritis or ankylosing spondylitis [13,24]. Moreover, IL-17a regulates the function of IL-22, which acts as a promoter of inflammation in the presence of IL-17a but contributes to the regulation of cell proliferation for repair following injury in the absence of IL-17a [17]. However, while we could detect increased IL-17a expression in the lung and right ventricular tissue of the PH rats, which was attenuated by treatment with MAC, and we also could show elevated serum levels of IL-17a in PH rats, IL17a could not be detected in the serum of PH patients. In contrast, the cytokine has been recently described to be upregulated in an immunophenotyping study using machine learning in pulmonary arterial hypertension patients [25].

Since the overall goal of this study was to identify a potential novel biomarker improving diagnosis of PH in patients, we now focus on the discussion of IL-22 results in the different clinical groups distinguished by current guidelines.

In general, IL-22 belongs to the IL-10 cytokine family and is produced by Th17 cells, but also by natural killer cells [26]. It is elevated in the context of lung damage due to allergies, infections or fibrosis [27] , as well as in chronic inflammatory and autoimmune diseases [28,29]. It has also been shown that IL-22 can promote the growth of smooth muscle cells and endothelial dysfunction and remodeling, which are known to be some of the pathophysiological changes that occur in the course of PH [28,30,31,32,33], irrespective of the etiology.

Few studies have shown possible causes and correlations for IL-22 elevation due to the different underlying diseases and triggers for PH in the past. In PH patients of group I, an infection is discussed as a possible trigger for the disease development in patients with genetic predisposition [32]. In this context, an immigration of neutrophils releasing proteolytic granules that result in tissue damage was shown and discussed as a possible starting point for the increased distribution of IL-22 [34,35] as a means to provide tissue homeostasis. Patients with PH group II suffer from left heart failure, which causes a massive volume overload in the pulmonary circulation that leads to PH of the post-capillary type [36]. An elevation of IL-22 in patients with PH group II could be based on the secondary effects on pulmonary circulation with mechanical damage of the endothelium. Studies concerning IL-22 in patients with PH group II have not existed up to now. Thus, in our current analysis, a significant elevation of the cytokine in this particular group has been shown for the first time.

In patients with COPD, a possible cause for PH group III, increased serum and sputum levels of IL-22 and also elevated levels in lung tissue were demonstrated in the study of Le Rouzic et al. [37]. Another study demonstrated the aggravating role of IL-22 on neutrophile inflammation and endothelial remodeling in patients with COPD [38].

For patients with deep vein thrombosis, elevated IL-22 levels could be shown [39], which is interesting regarding PH group IV. Also, in patients with chronic thromboembolic pulmonary hypertension (CTEPH), different cytokines showed increased levels, such as IL-1β, IL-2, IL-4, IL-6, IL-8 and IL-10 [40,41]. As a member of the IL-10 family, IL-22 is also conceivable as a valid biomarker for CTEPH, as it was elevated in our PH group IV collective as well.

Since IL-22 was elevated in all PH patient groups compared to controls in our study, the biomarker will be possibly helpful for early detection of the disease in general, but will currently not contribute to etiological classification with respect to the different PH groups. However, if a patient presents with a known disease commonly leading to PH at later stages, e.g., left heart or lung disease as well as chronic thromboembolism, the inclusion of IL-22 as additional biomarker for therapy surveillance could be useful to detect PH and consecutive right heart failure development.

To the best of our knowledge, this is the first study to correlate clinical parameters from PH patients of different clinical groups with IL-22 serum levels. By doing so, we can demonstrate multiple positive correlations, e.g., regarding right ventricular diameter, right atrial area, systolic PAP determined by echocardiography and invasively measured PAP, as well as BNP. TAPSE and the 6-min-walking test showed a negative correlation with IL-22 serum levels. Moreover, there were significantly different IL-22 serum levels in the comparison of the functional classes patients presented with.

The 6 min walking test is a simple, reproducible test for patients with PH that is strongly correlated with mortality [42]. Our results regarding the negative correlation of IL-22 with the 6 min walking distance are consistent with findings for other biomarkers such as CRP [43], Osteopontin [44], BNP [45], Tenascin-C [46] and CT-pro-ET-1 [47], which are all negatively correlated with the 6 min walking distance in PH patients.

BNP and NT-proBNP, respectively, are currently the only biomarkers for PH which are used in daily clinical practice [1,48] , but they are not specific for this disease. However, there are multiple studies that show significant correlations between BNP/NT-proBNP and the 6 min walking test, invasive PAP, right atrial pressure or the functional class [45,49,50]. Our data not only show a positive correlation between BNP and IL-22 for patients with PH but further positive correlations between IL-22 and different echocardiographic, laboratory and invasive parameters. In addition, in our study, IL-22 levels increase with a higher functional class as a means of increasing disease severity. Similarly, for patients with psoriasis, a correlation between increasing IL-22 levels and increased disease severity was shown [51].

Our data show significant correlations between IL-22 levels and echocardiographic parameters, such as systolic PAP or right ventricular diameter, that are also known to reflect the severity of PH [52,53]. Because of the disturbed hemodynamic in patients with PH, a remodeling of the right ventricle takes place with a proliferation of smooth muscle tissue, for which the involvement of IL-22 has already been shown [30]. A different possibility would be an inflammatory reaction of the myocardium due to mechanic strain because of the elevated afterload [10,54].

Our results also show a positive correlation between IL-22 and the invasive measured mPAP, which is of fundamental importance in the diagnostic of PH. It is also a validated predictor for the progression of disease, mortality and treatment response [48,55]. Another very important hemodynamic value is the PVR, which is used to differentiate pre-capillary PH and is also correlated to the disease severity and mortality [52,56,57]. Here, we could show a significant positive correlation with IL-22 serum levels as well.

To date, other studies could not demonstrate a correlation of PVR and mPAP with multiple cytokines, such as IL-1β, IL-2, IL-4, IL-5, IL-6, IL-8, IL-10, IL-12, IL-13, Interferon-ɣ and TNF-⍺ [40,58]. In this regard, IL-22 is a promising novel biomarker as it is widely correlated with a multitude of hemodynamic and functional disease markers in patients with PH in our study.

Based upon this pre-existing data and our own results, we could show that elevated serum IL-22 could be a marker not only for the existence of PH but also for the severity of the disease and so could serve not only as a diagnostic marker but as one with prognostic value as well.

Limitations for this study include the relatively small number of subjects, especially in the human collective. Baseline characteristics between the human PH and control group partly showed significant differences. Patients in the PH group had more co-morbidities, which could lead to greater differences in specific clinical markers and even different IL-22 serum levels. Further studies with larger patient collectives are therefore needed. Also, there are no follow-up data on the human PH patients regarding serum IL-22 levels and clinical outcome parameters after treatment of PH. Thus, the decrease in IL-22 levels after treatment of PH shown in the rat model could not be examined in the human cohort yet. Of course, this is an important subject for future studies with PH patients. Moreover, a direct comparison of IL-22 with other biomarkers in the same collective would be of great interest to address in that context. Taken together, further research addressing these limitations is required before definitive conclusions can be drawn regarding the clinical utility of IL-22 as a biomarker in PH.

## 5. Conclusions

Serum IL-22 was 4.6 times higher in the PH patient collective than in the control collective. It was elevated regardless of the PH etiology across all PH groups. Correlations between serum IL-22 and a multitude of hemodynamic, laboratory, echocardiographic and functional markers could be demonstrated. Based upon our results, we propose the use of serum IL-22 as a novel biomarker for diagnosis of PH, as there are currently no disease-specific biomarkers. Regarding the promising results of the animal model, serum IL-22 maybe even used for therapeutic surveillance in patients with PH. Further studies in larger collectives regarding these findings are necessary.

## Figures and Tables

**Figure 1 ijms-25-03985-f001:**
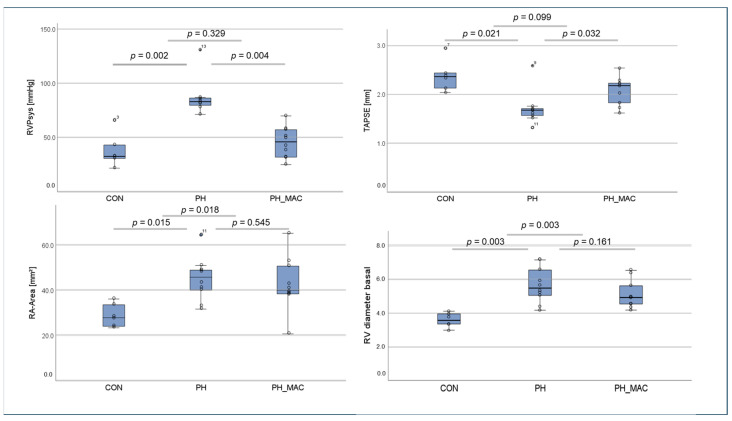
Differences in RVPsys, TAPSE, RA area and basal RV diameter measured by echocardiography in the experimental rat model groups (n = 26, CON n = 6, PH n = 10, PH_MAC n = 10) on day 27 and day 28, respectively, compared using the Mann–Whitney-U-test.

**Figure 2 ijms-25-03985-f002:**
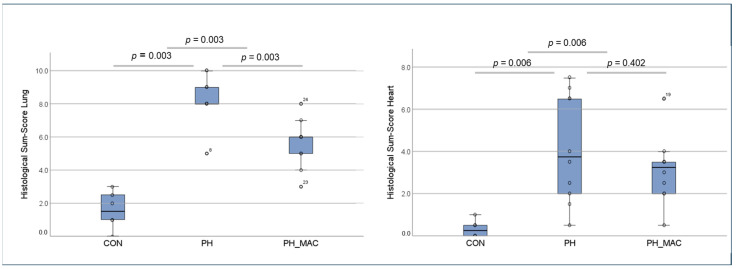
Histological sum-scores representing lung tissue damage (**left**) and right ventricular tissue (**right**) damage among the experimental rat groups (n = 26, CON n = 6, PH n = 10, PH_MAC n = 10), compared by Mann–Whitney-U-test.

**Figure 3 ijms-25-03985-f003:**
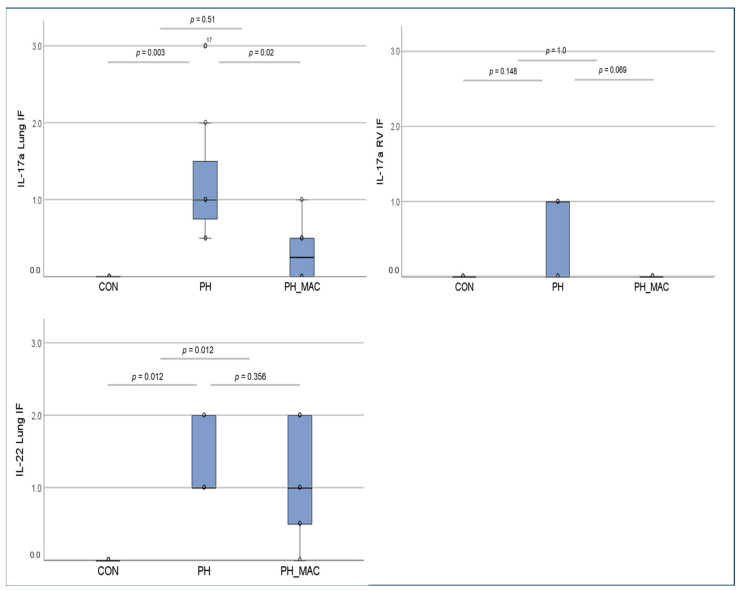
Immunofluorescence-based detection and semi-quantitative scoring representing the IF of IL-17a in the lung and in RV, as well as IL-22 in the lung, in the different rat groups (n = 26, CON n = 6, PH n = 10, PH_MAC n = 10), compared using Mann–Whitney-U-test.

**Figure 4 ijms-25-03985-f004:**
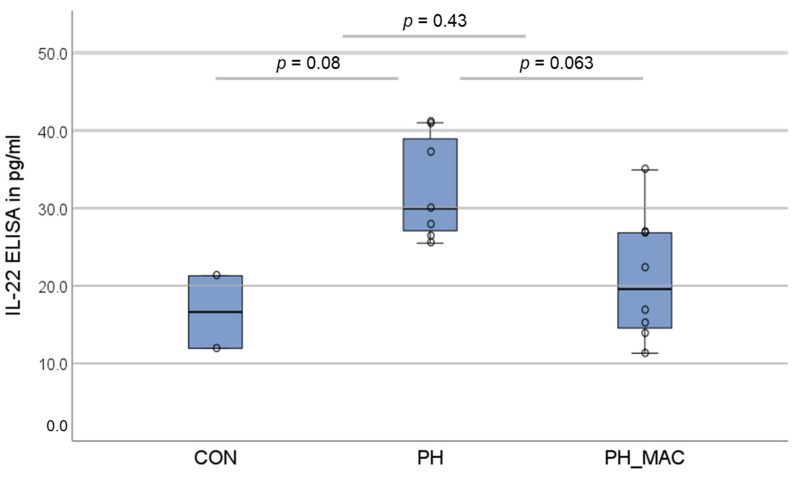
Serum concentrations of IL-22 determined using ELISA in the different rat groups (n = 26, CON n = 6, PH n = 10, PH_MAC n = 10), compared using the Mann–Whitney-U-test.

**Figure 5 ijms-25-03985-f005:**
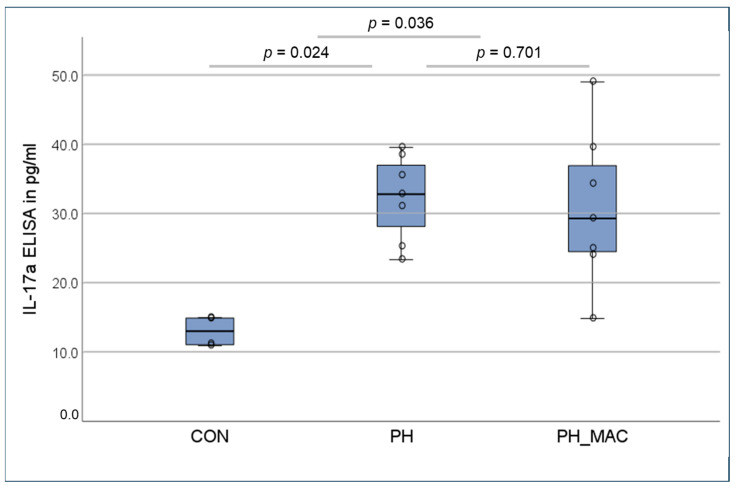
Serum concentrations of IL-17a determined using ELISA in the different rat groups (n = 26, CON n = 6, PH n = 10, PH_MAC n = 10), compared using the Mann–Whitney-U-test.

**Figure 6 ijms-25-03985-f006:**
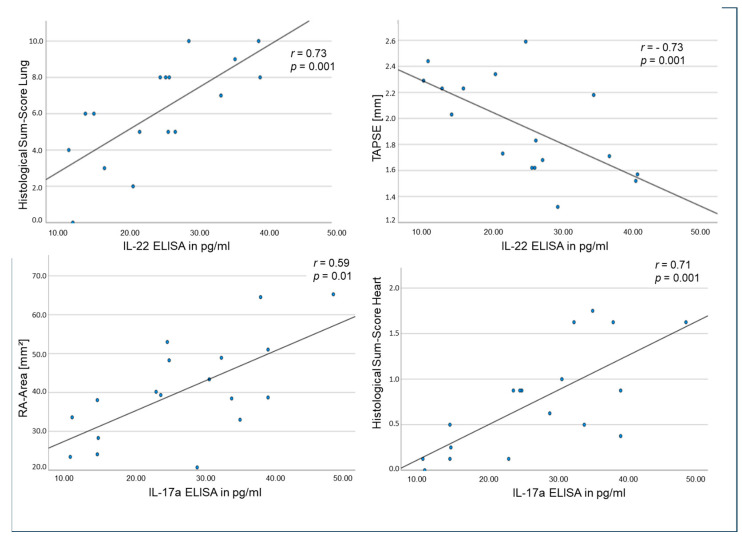
Correlations of serum IL-22 levels with histological lung tissue damage (**upper left**) and TAPSE (**upper right**) as well as of serum IL-17a levels with right atrial area in echocardiography (**down left**) and histological sum-score in RV (**down right**) using Spearman’s rank correlation.

**Figure 7 ijms-25-03985-f007:**
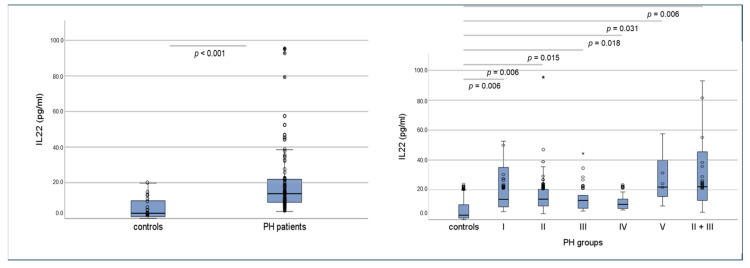
Serum levels of IL-22 in human PH patients, PH subgroups and controls, using the Mann–Whitney-U-test. * = outlying value.

**Figure 8 ijms-25-03985-f008:**
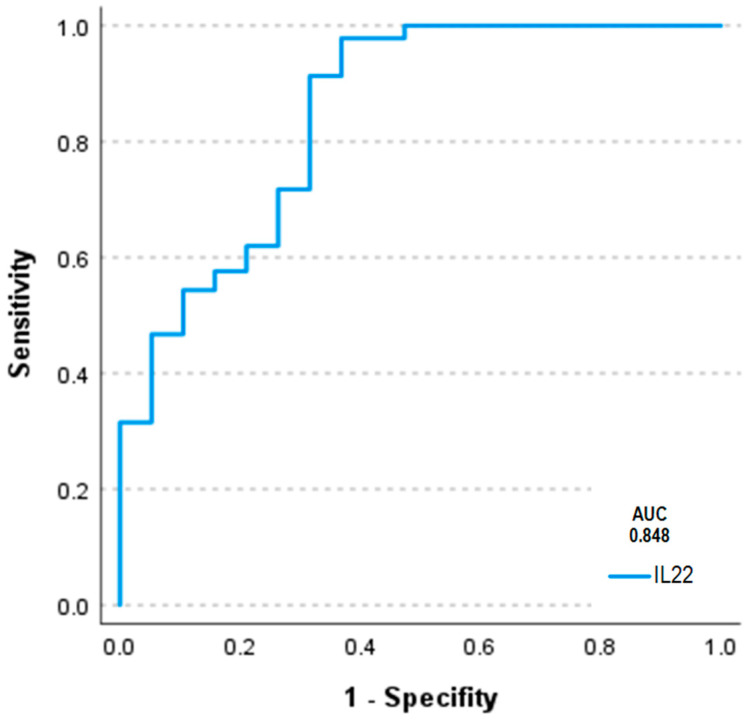
ROC analysis for the discrimination of patients with and without PH revealing an AUC for IL-22 of 0.848.

**Table 1 ijms-25-03985-t001:** Significant correlations between echocardiographic/hemodynamic/histologic parameters and ELISA results using Spearman’s rank correlation.

Immunohistochemical/ ELISA Result	Echocardiographic/Hemodynamic/ Histologic Parameter	r-Value	*p*-Value
IL-17a ELISA	Histological Sum-Score Lung	0.580	0.012
IL-17a ELISA	Histological Sum-Score Heart	0.711	0.001
IL-17a ELISA	RA-Area [mm^2^]	0.592	0.010
IL-17a ELISA	TAPSE [mm]	−0.535	0.022
IL-17a ELISA	RVOT [mm]	0.696	0.001
IL-22 ELISA	Histological Sum-Score Lung	0.731	0.001
IL-22 ELISA	Histological Sum-Score Heart	0.497	0.042
IL-22 ELISA	RV-Catheter RVPsys [mmHg]	0.550	0.034
IL-22 ELISA	TAPSE [mm]	−0.730	0.001

**Table 2 ijms-25-03985-t002:** Baseline characteristics of PH patients and the control group.

	Control Group (n = 30)	PH Patients (n = 92)	*p*-Value
**Clinical parameters**			
Age (years)	65 ± 7	75 ± 12	<0.001
Male gender (%)	37	30	n.s.
BMI (kg/m^2^)	27.4 ± 4.3	29.5 ± 6.7	n.s.
Systolic BP (mmHg)	153.6 ± 28.4	141.9 ± 27.0	0.026
Diastolic BP (mmHg)	87.6 ± 12.6	79.5 ± 14.5	0.011
Heart rate (beats per min)	70.3 ± 11.5	76.7 ± 13.8	0.026
Functional class	1.4 ± 0.5 (NYHA)	2.4 ± 0.8	<0.001
**Comorbidities**			
Arterial hypertension (%)	87	86	n.s.
Coronary artery disease (%)	10	28	0.042
Hypertensive heart disease (%)	66	63	n.s.
Atrial fibrillation (%)	13	50	<0.001
Lung diseases (%)	17	52	0.001
COPD (%)	3	23	0.015
Pulmonary fibrosis (%)	0	14	0.029
Chronic kidney disease (GFR < 50 mL/min) (%)	3	55	<0.001
Hyperlipidemia (%)	63	61	n.s.
Diabetes mellitus (%)	13	44	0.003
Obesity (BMI > 30 kg/m^2^) (%)	33	48	n.s.
Autoimmune diseases (%)	0	21	0.007
Smoking (%)	37	34	n.s.
**Medication (%)**			
ASA(%)	40	20	0.031
Beta blocker (%)	53	65	n.s.
ACE inhibitor/Sartan (%)	73	62	n.s.
Calcium channel blocker (%)	37	33	n.s.
Diuretic (%)	33	83	<0.001
Statin (%)	50	66	n.s.
Prednisolone (%)	7	14	n.s.
Inhaled glucocorticoid (%)	13	20	n.s.
**Laboratory parameters**			
BNP (pg/mL)	38 ± 33.1	199.0 ± 782.5	<0.001
CRP (mg/L)	2.0 ± 4.6	5.7 ± 17.1	<0.001
Creatinine (µmol/L)	75.5 ± 10.1	104.0 ± 78.7	<0.001
LDL (mmol/L)	3.7 ± 1.4	2.4 ± 1.0	0.003
Hemoglobin (mmol/L)	8.6 ± 0.8	7.9 ± 1.2	0.001
Leukocytes (Gpt/L)	6.9 ± 2.3	7.0 ± 2.3	n.s.
Platelets (Gpt/L)	258 ± 51.8	223.0 ± 67.0	0.006

Data presented as mean ± standard deviation or percentage. Laboratory data presented as median ± standard deviation. Abbreviations: ACE: angiotensin-converting enzyme, ASA: acetylsalicylic acid, BMI: body mass index, BNP: brain natriuretic peptide, BP = blood pressure, COPD: chronic obstructive pulmonary disease, CRP: C-reactive protein, GFR: glomerular filtration rate, LDL: low-density lipoprotein, n: number, n.s.: not significant, PH: pulmonary hypertension.

**Table 3 ijms-25-03985-t003:** Echocardiographic parameters of the control group, the PH group and the PH subgroups.

Parameter	Control Group	PH_all_groups_	*p*-Value *	PH l	PH ll	PH lll	PH lV	PH V	PH ll/lll	*p*-Value **
PAPsys (mmHg)	25.5 ± 4.1	52 ± 17.8	<0.001	53.2 ± 19.9	51.6 ± 15.7	55.5 ± 17.7	37.8 ± 20.1	35.0 ± 6.2	57.4 ± 17.1	n.s.
TAPSE (mm)	24.7 ± 4.0	18 ± 5.3	<0.001	17.6 ± 6.6	16.7 ± 4.2	20.1 ± 6.6	19.2 ± 4.3	(32; n = 1)	16.0 ± 3.6	n.s.
RAA (cm^2^)	16 ± 2.6	26 ± 10.7	<0.001	23 ± 10.9	29.6 ± 11.1	22.3 ± 7.4	18.7 ± 5.3	(16; n = 1)	30.3 ± 11.3	0.03
Relevant Vitium (%)	3	66	0.001	77	87	11	44	33	65	0.001
IVSd (mm)	11.6 ± 2.5	13 ± 9.2	n.s.	11.8 ± 3.2	12.4 ± 3.0	13.8 ± 3.0	11.4 ± 2.6	(10; n = 1)	12.0 ± 2.3	n.s.
LVEF (%)	65.2 ± 9.5	59 ± 11	0.015	59.9 ± 8.8	56.3 ± 12.9	62.6 ± 8.2	63.3 ± 5.6	70 ± 6.4	56.2 ± 11.4	n.s.

Data presented as mean ± standard deviation. * *p*-value between controls and PH patients; ** *p*-value between the different PH groups. Abbreviations: IVSd: interventricular septal end diastole, PAPsys: systolic pulmonary artery pressure, RAA: right atrial area, LVEF: left ventricular ejection fraction, PH: pulmonary hypertension. n.s. is the abbrevation for “not significant”.

## Data Availability

All data reported in this article are available in the article or in its Appendix A.

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
