# Peer review of "Identification of Serum Interleukin-22 as Novel Biomarker in Pulmonary Hypertension: A Translational Study"

_ijms, 2024, doi:10.3390/ijms25073985_

Round 1

Reviewer 1 Report

Comments and Suggestions for Authors

This article by Dinesh and colleagues explores IL-17a and IL-22 expression as potential biomarkers for pulmonary hypertension (PH) using a rat model and serum analysis in PH patients. The authors report that IL-22 expression increased significantly in rat lungs, correlating with elevated serum levels in both rat and human PH groups. Finally, the authors propose IL-22, as an inflammatory biomarker for PH.

In general, the article is well-written, but, a lot of the figures feel redundant and could be grouped in panels. My biggest problem with the article is that the claims on the conclusions are too big and the data is too limited and not strong enough for them. I have several points to help improve the article:

Major points:

-       In general, and for all the figures, the authors should add all the data points, and the figure legends should be informative enough. So please, add the n number for the plots, and at least the statistical test used.

-       When comparing multiple groups authors should use the appropriate multiple-comparison correction.

-       There is not a single explanation of what MAC represents until line 431, please put the whole name of the drug you are using the first time you write the acronym.

-       Line 112, avoid driving the readers through a rabbit hole of publications. Briefly explain the method or explain it extensively in the methods section. Figure 2 is useless if there are no explanations of the method. I would also advise adding representative images of each condition. If the figure is not going to add anything new that has not been reported in the literature just put it as supporting information that your model worked (showing vascularization of the pulmonary vessels for example).

-       Figures 3, 4, and 5 are terrible, an immunofluorescence showing only nuclei and the ILx is not informative at all, the authors could be showing us any other tissue. Please add markers for lung cells, immune cells, and cardiac cells (whatever is relevant for the sample). Show where the production of the ILs is happening. Also, the control has a signal for the ILs, even if it is a little expression, it is not 0 like the authors plot it. Confocal imaging would be recommended if the plan is to quantify.

-       I would advise doing some quantification of the number of T helper and NK cells in the lung homogenate (flow cytometry would be the best if the authors want to relate it to the IL production).

-       The description of the control samples used for the ELISA was quite hard to find. Recruiting random people in their hospital is a way of calling for some extra variability, I would not call them healthy controls given the amount of comorbidities. If they could use actual healthy donors (people donating blood for example), it would be better. If they are controls, why do they have a functional class?

-       If the authors want to estate that IL-22 could be used for diagnosis they should do at least a linear regression and show ROC curves, the number is enough to show preliminarily how well they can distinguish patients and controls.

Minor points:

-       Line 58 has a very out-of-pocket self-citation that does not add anything to the current article.

-       Figures 8-9 could be mixed into one.

-       Figure 11 should be supplementary.

-       Figure 12 shows an increase in IL-22 which is something somehow expected as disease progresses, it should not be a standalone figure.

-       There are other articles analyzing the immune profile in plasma from PAH patients, and there is no mention in the discussion (PMID: 30661465 discusses IL-17 a bit).

Author Response

This article by Dinesh and colleagues explores IL-17a and IL-22 expression as potential biomarkers for pulmonary hypertension (PH) using a rat model and serum analysis in PH patients. The authors report that IL-22 expression increased significantly in rat lungs, correlating with elevated serum levels in both rat and human PH groups. Finally, the authors propose IL-22, as an inflammatory biomarker for PH.

In general, the article is well-written, but, a lot of the figures feel redundant and could be grouped in panels. My biggest problem with the article is that the claims on the conclusions are too big and the data is too limited and not strong enough for them. I have several points to help improve the article:

General response: The authors are extremely grateful for this constructive review and for giving a variety of very important advices. We tried to realize the suggestions of the reviewer whenever possible and could thereby significantly improve the presentation quality of our study. Please find a point-by-point response below.

Major points:

-       In general, and for all the figures, the authors should add all the data points, and the figure legends should be informative enough. So please, add the n number for the plots, and at least the statistical test used.

Response: Many thanks for that important suggestion. We have now added data points to all figures throughout the manuscript and improved the figure legends by giving detailed information regarding n number and statistical test used. The changes can be found in the revised version of the manuscript (track changes mode).

-       When comparing multiple groups authors should use the appropriate multiple-comparison correction.

Response: The reviewer is completely right. We used Bonferroni-Holm correction method for multiple-comparison correction and updated the p values throughout the manuscript. Moreover this method has been added the material and methods chapter (revised version, track changes mode). 

-       There is not a single explanation of what MAC represents until line 431, please put the whole name of the drug you are using the first time you write the acronym.

Response: The authors are very sorry for that inconvenience. We have explained MAC in the revised version of the manuscript at the place of first use of the abbreviation (see track changes mode).

-       Line 112, avoid driving the readers through a rabbit hole of publications. Briefly explain the method or explain it extensively in the methods section. Figure 2 is useless if there are no explanations of the method. I would also advise adding representative images of each condition. If the figure is not going to add anything new that has not been reported in the literature just put it as supporting information that your model worked (showing vascularization of the pulmonary vessels for example).

Response: We fully agree with the criticism of the reviewer. We nor have explained the method in details in the material and methods section of the revised version of the manuscript. As we noticed, the manuscript file version that was send out for review did not contain the material and methods part for unknown reasons. We really hope that the questions addressed by the reviewer is now answered in a satisfying way.

-       Figures 3, 4, and 5 are terrible, an immunofluorescence showing only nuclei and the ILx is not informative at all, the authors could be showing us any other tissue. Please add markers for lung cells, immune cells, and cardiac cells (whatever is relevant for the sample). Show where the production of the ILs is happening. Also, the control has a signal for the ILs, even if it is a little expression, it is not 0 like the authors plot it. Confocal imaging would be recommended if the plan is to quantify.

Response: Many thanks for pointing out that relevant problem. Of course, it would have been nice and much more convincing, if we had added further IHC markers, e.g., for lung, immune or cardiac cells. Honestly and unfortunately, this has not been done in detail and cannot be added afterwards due to the non-availability of tissue, which has been fully consumed in the meantime. To overcome this problem, we decided to remove the images from the revised version of the manuscript. We really hope that the reviewer will accept our decision. The “0” is derived from the semiquantitative method and, of course, includes also very low expression levels occurring in controls. For clarification, we added “0=nearly no expression” in the material an methods chapter of the revised version of the manuscript.

A useful software-based quantification was not possible due to the heterogeneity of samples and we had to decide for a semiquantitative method performed by experienced scientists.  

-       I would advise doing some quantification of the number of T helper and NK cells in the lung homogenate (flow cytometry would be the best if the authors want to relate it to the IL production).

Response: The authors are grateful for this suggestions and will try to add this method in further studies. Retrospectively for this study, we cannot do so since there is not tissue left for performing flow cytometry.

-       The description of the control samples used for the ELISA was quite hard to find. Recruiting random people in their hospital is a way of calling for some extra variability, I would not call them healthy controls given the amount of comorbidities. If they could use actual healthy donors (people donating blood for example), it would be better. If they are controls, why do they have a functional class?

Response: The reviewer is completely right that the description of the control collective was insufficient. We have now clarified in the material and methods section of the revised version of the manuscript: we used cardiovascular risk patients not showing evidence pf pulmonary hypertension, ischemic or structural heart disease. All patients underwent coronary angiography to exclude coronary artery disease and detailed transthoracic echocardiography. Moreover, chronic inflammatory, autoimmune or neoplastic diseases were excluded. The functional class in the control group represents the NYHA functional class, this important information is now given in table 2 of the revised version of the manuscript.

-       If the authors want to estate that IL-22 could be used for diagnosis they should do at least a linear regression and show ROC curves, the number is enough to show preliminarily how well they can distinguish patients and controls.

Response: This is a very helpful idea and we performed ROC analysis as suggested and added these very important information in the revised version of the manuscript.

Minor points:

-       Line 58 has a very out-of-pocket self-citation that does not add anything to the current article.

Response: The reviewer is right that this self-citation seems to be “out-of-pocket” at first glance. But, this study gave the motivational impetus to look for the interleukins reported in our current study presented here, since the pathway-focused gene expression analyses could identify them as potentially interesting. Thus, the authors strongly believe that it is necessary to leave this citation in. We really hope that the reviewer will follow our explanation.

-       Figures 8-9 could be mixed into one.

Response: That’s completely right and we did mix it into one figure.

-       Figure 11 should be supplementary.

Response: Done.

-       Figure 12 shows an increase in IL-22 which is something somehow expected as disease progresses, it should not be a standalone figure.

Response: The figure has been transferred to supplements as suggested by the reviewer.

-       There are other articles analyzing the immune profile in plasma from PAH patients, and there is no mention in the discussion (PMID: 30661465 discusses IL-17 a bit).

Response: The reference has been added in the discussion chapter of the revised version of the manuscript.

This article by Dinesh and colleagues explores IL-17a and IL-22 expression as potential biomarkers for pulmonary hypertension (PH) using a rat model and serum analysis in PH patients. The authors report that IL-22 expression increased significantly in rat lungs, correlating with elevated serum levels in both rat and human PH groups. Finally, the authors propose IL-22, as an inflammatory biomarker for PH.

In general, the article is well-written, but, a lot of the figures feel redundant and could be grouped in panels. My biggest problem with the article is that the claims on the conclusions are too big and the data is too limited and not strong enough for them. I have several points to help improve the article:

General response: The authors are extremely grateful for this constructive review and for giving a variety of very important advices. We tried to realize the suggestions of the reviewer whenever possible and could thereby significantly improve the presentation quality of our study. Please find a point-by-point response below.

Major points:

-       In general, and for all the figures, the authors should add all the data points, and the figure legends should be informative enough. So please, add the n number for the plots, and at least the statistical test used.

Response: Many thanks for that important suggestion. We have now added data points to all figures throughout the manuscript and improved the figure legends by giving detailed information regarding n number and statistical test used. The changes can be found in the revised version of the manuscript (track changes mode).

-       When comparing multiple groups authors should use the appropriate multiple-comparison correction.

Response: The reviewer is completely right. We used Bonferroni-Holm correction method for multiple-comparison correction and updated the p values throughout the manuscript. Moreover this method has been added the material and methods chapter (revised version, track changes mode). 

-       There is not a single explanation of what MAC represents until line 431, please put the whole name of the drug you are using the first time you write the acronym.

Response: The authors are very sorry for that inconvenience. We have explained MAC in the revised version of the manuscript at the place of first use of the abbreviation (see track changes mode).

-       Line 112, avoid driving the readers through a rabbit hole of publications. Briefly explain the method or explain it extensively in the methods section. Figure 2 is useless if there are no explanations of the method. I would also advise adding representative images of each condition. If the figure is not going to add anything new that has not been reported in the literature just put it as supporting information that your model worked (showing vascularization of the pulmonary vessels for example).

Response: We fully agree with the criticism of the reviewer. We nor have explained the method in details in the material and methods section of the revised version of the manuscript. As we noticed, the manuscript file version that was send out for review did not contain the material and methods part for unknown reasons. We really hope that the questions addressed by the reviewer is now answered in a satisfying way.

-       Figures 3, 4, and 5 are terrible, an immunofluorescence showing only nuclei and the ILx is not informative at all, the authors could be showing us any other tissue. Please add markers for lung cells, immune cells, and cardiac cells (whatever is relevant for the sample). Show where the production of the ILs is happening. Also, the control has a signal for the ILs, even if it is a little expression, it is not 0 like the authors plot it. Confocal imaging would be recommended if the plan is to quantify.

Response: Many thanks for pointing out that relevant problem. Of course, it would have been nice and much more convincing, if we had added further IHC markers, e.g., for lung, immune or cardiac cells. Honestly and unfortunately, this has not been done in detail and cannot be added afterwards due to the non-availability of tissue, which has been fully consumed in the meantime. To overcome this problem, we decided to remove the images from the revised version of the manuscript. We really hope that the reviewer will accept our decision. The “0” is derived from the semiquantitative method and, of course, includes also very low expression levels occurring in controls. For clarification, we added “0=nearly no expression” in the material an methods chapter of the revised version of the manuscript.

A useful software-based quantification was not possible due to the heterogeneity of samples and we had to decide for a semiquantitative method performed by experienced scientists.  

-       I would advise doing some quantification of the number of T helper and NK cells in the lung homogenate (flow cytometry would be the best if the authors want to relate it to the IL production).

Response: The authors are grateful for this suggestions and will try to add this method in further studies. Retrospectively for this study, we cannot do so since there is not tissue left for performing flow cytometry.

-       The description of the control samples used for the ELISA was quite hard to find. Recruiting random people in their hospital is a way of calling for some extra variability, I would not call them healthy controls given the amount of comorbidities. If they could use actual healthy donors (people donating blood for example), it would be better. If they are controls, why do they have a functional class?

Response: The reviewer is completely right that the description of the control collective was insufficient. We have now clarified in the material and methods section of the revised version of the manuscript: we used cardiovascular risk patients not showing evidence pf pulmonary hypertension, ischemic or structural heart disease. All patients underwent coronary angiography to exclude coronary artery disease and detailed transthoracic echocardiography. Moreover, chronic inflammatory, autoimmune or neoplastic diseases were excluded. The functional class in the control group represents the NYHA functional class, this important information is now given in table 2 of the revised version of the manuscript.

-       If the authors want to estate that IL-22 could be used for diagnosis they should do at least a linear regression and show ROC curves, the number is enough to show preliminarily how well they can distinguish patients and controls.

Response: This is a very helpful idea and we performed ROC analysis as suggested and added these very important information in the revised version of the manuscript.

Minor points:

-       Line 58 has a very out-of-pocket self-citation that does not add anything to the current article.

Response: The reviewer is right that this self-citation seems to be “out-of-pocket” at first glance. But, this study gave the motivational impetus to look for the interleukins reported in our current study presented here, since the pathway-focused gene expression analyses could identify them as potentially interesting. Thus, the authors strongly believe that it is necessary to leave this citation in. We really hope that the reviewer will follow our explanation.

-       Figures 8-9 could be mixed into one.

Response: That’s completely right and we did mix it into one figure.

-       Figure 11 should be supplementary.

Response: Done.

-       Figure 12 shows an increase in IL-22 which is something somehow expected as disease progresses, it should not be a standalone figure.

Response: The figure has been transferred to supplements as suggested by the reviewer.

-       There are other articles analyzing the immune profile in plasma from PAH patients, and there is no mention in the discussion (PMID: 30661465 discusses IL-17 a bit).

Response: The reference has been added in the discussion chapter of the revised version of the manuscript.

Reviewer 2 Report

Comments and Suggestions for Authors

The study titled "Identification of serum Interleukin-22 as novel biomarker in pulmonary hypertension: a translational study" presents interesting findings regarding the identification of serum IL-22 as a potential novel biomarker in pulmonary hypertension (PH). The experiments conducted in rat models are commendably performed and provide robust and interesting results. However, the translational aspect of the study, particularly its application to humans, presents significant challenges that undermine the scientific integrity of the article.

While the rat model experiments are well designed and provide compelling evidence, the human cohort analysis suffers from several biases, as highlighted by the authors in the limitations section. In particular, the relatively small sample size of the human cohort, especially compared to the rat model, is a major limitation. This limitation raises concerns about the generalisability of the findings to the wider human population affected by PH.

In addition, the observed differences in baseline characteristics between the PH and control groups in the human cohort introduce confounding variables that may bias the interpretation of the results. The presence of comorbidities in the PH group, which were not adequately controlled for in the study, could influence certain clinical markers and potentially confound the observed serum IL-22 levels.

In addition, the lack of follow-up data on the human PH patients regarding serum IL-22 levels and clinical outcome parameters after treatment further reduces the reliability of the findings. The lack of longitudinal data precludes an assessment of the relationship between IL-22 levels and disease progression or treatment response in human patients.

The authors appropriately acknowledge these limitations and the need for further studies with larger patient cohorts to validate their findings.

However, the current state of the study, particularly the human cohort analysis, renders the article scientifically unsound and casts doubt on the validity of the conclusions drawn.

In conclusion, although the findings of the study regarding IL-22 as a potential biomarker in PH are intriguing, the limitations associated with the human cohort analysis significantly undermine the credibility of the results. Further research addressing these limitations is required before definitive conclusions can be drawn regarding the clinical utility of IL-22 as a biomarker in PH.

For these reasons I suggest to reconsider the article after major revision

Author Response

The study titled "Identification of serum Interleukin-22 as novel biomarker in pulmonary hypertension: a translational study" presents interesting findings regarding the identification of serum IL-22 as a potential novel biomarker in pulmonary hypertension (PH). The experiments conducted in rat models are commendably performed and provide robust and interesting results. However, the translational aspect of the study, particularly its application to humans, presents significant challenges that undermine the scientific integrity of the article.

Response: The authors are very grateful for that constructive and positive review and, of course, the human part of the study is somehow preliminary and has to be extended in further studies. Nevertheless, the authors believe that the main finding, the identification of IL-22 as potential novel biomarker or even therapeutic target in PH, is worth reporting at the current state of knowledge. Please find below a point-by-point response to all comments and suggestions, which we tried to address in the revised version of the manuscript.

While the rat model experiments are well designed and provide compelling evidence, the human cohort analysis suffers from several biases, as highlighted by the authors in the limitations section. In particular, the relatively small sample size of the human cohort, especially compared to the rat model, is a major limitation. This limitation raises concerns about the generalisability of the findings to the wider human population affected by PH.

Response: This is an assessment, which is completely right also in the eyes of the authors. Nevertheless, as stated above, the results should be reported now to give an impetus to perform human IL-22 studies at larger, well characterized patient cohorts and to also include longitudinal measurements to study correlations to disease progression / regression or even therapy response (therapy surveillance marker?). This will be done by the authors and, hopefully, also others groups facilitating fast knowledge gain in the interest of our patients.

In addition, the observed differences in baseline characteristics between the PH and control groups in the human cohort introduce confounding variables that may bias the interpretation of the results. The presence of comorbidities in the PH group, which were not adequately controlled for in the study, could influence certain clinical markers and potentially confound the observed serum IL-22 levels.

Response: The reviewer is completely right and the criticism may be due to the imperfect description of the control collective in the current version of the manuscript. We have now clarified in the material and methods section of the revised version of the manuscript: we used cardiovascular risk patients not showing evidence pf pulmonary hypertension, ischemic or structural heart disease. All patients underwent coronary angiography to exclude coronary artery disease and detailed transthoracic echocardiography. Moreover, chronic inflammatory, autoimmune or neoplastic diseases were strictly excluded. Thus, we tried our very best to reduce potential confounders regarding cytokine release due to comorbidities. See improved description of the controls group in the material and methods sections of the revised version of the manuscript.

In addition, the lack of follow-up data on the human PH patients regarding serum IL-22 levels and clinical outcome parameters after treatment further reduces the reliability of the findings. The lack of longitudinal data precludes an assessment of the relationship between IL-22 levels and disease progression or treatment response in human patients.

Response: This is, again, completely correct. We would like to refer to response 2.

The authors appropriately acknowledge these limitations and the need for further studies with larger patient cohorts to validate their findings. However, the current state of the study, particularly the human cohort analysis, renders the article scientifically unsound and casts doubt on the validity of the conclusions drawn.

Response: Many thanks for acknowledging that we clearly stated the limitations of our study. Nevertheless, as already stated above, we really hope to contribute an impetus for further research in the field.

In conclusion, although the findings of the study regarding IL-22 as a potential biomarker in PH are intriguing, the limitations associated with the human cohort analysis significantly undermine the credibility of the results. Further research addressing these limitations is required before definitive conclusions can be drawn regarding the clinical utility of IL-22 as a biomarker in PH.

Response: The authors are very grateful for summarizing the study limitation precisely. We have included a further statement in the limitations chapter of the revised version of the manuscript and hope that the reviewer will allow us to use his statement with minor modifications?

For these reasons I suggest to reconsider the article after major revision

Response: Thank your very much. Also integrating the comments and suggestions of another reviewer, we generated a major revision of our study and really hope that the manuscript will now be suitable for publication in the International Journal of Molecular Sciences.

Round 2

Reviewer 1 Report

Comments and Suggestions for Authors

I am glad to see that most of my comments were taken into account, I think the manuscript has improved in terms of its claims and what data was supporting.

- Taking out the Immunofluorescences was a good point if you are unable to repeat them with proper markers, it is a pity that the authors did not save some tissue in paraffin taking into account they were planning to submit an article.

Major points:

- The description of how the linear regression was performed is too simple, please explain it in detail, did the authors split the cohort in discovery and validation? or did they just put everything together and run the model? 

Author Response

I am glad to see that most of my comments were taken into account, I think the manuscript has improved in terms of its claims and what data was supporting.

Response: Many thanks for giving such a positive feed back. The authors are glad that most reviewer comments could be fulfilled satisfactory and are grateful for all the support of the reviewer, which helped to increase the quality of the manuscript.

- Taking out the Immunofluorescences was a good point if you are unable to repeat them with proper markers, it is a pity that the authors did not save some tissue in paraffin taking into account they were planning to submit an article.

Response: The reviewer is completely right. We are sorry and will keep the criticism in mind for further studies. Thank you for understanding. 

Major points:

- The description of how the linear regression was performed is too simple, please explain it in detail, did the authors split the cohort in discovery and validation? or did they just put everything together and run the model? 

Response: Many thanks for that important question. Due to the relatively small study cohort, we decided no to split into a discovery and validation and just put everything together. We have stated this more clearer in the methods section of the revised version of the manuscript (see track changes mode). 

Reviewer 2 Report

Comments and Suggestions for Authors

The authors have responded thoroughly to all the comments made, and the changes in the text of the paper have made the human subjects part clearer. I therefore recommend that the paper be accepted as it stands.

Author Response

The authors have responded thoroughly to all the comments made, and the changes in the text of the paper have made the human subjects part clearer. I therefore recommend that the paper be accepted as it stands.

Response: Many thanks for giving such a positive feedback. The authors are grateful for the efforts the reviewer made to improve the quality of the manuscript.